# Synergistic 2D/3D Convolutional Neural Network for Hyperspectral Image Classification

**Xiaofei Yang [1,2]** **, Xiaofeng Zhang [1,2,\*], Yunming Ye [1,2,\*], Raymond Y. K. Lau [3]** **, Shijian Lu [4],**
**Xutao Li [1,2] and Xiaohui Huang [5]**

[1] Harbin Institute of Technology, Shenzhen 518055, China; yangxiaofei@stu.hit.edu.cn (X.Y.);
   lixutao@hit.edu.cn (X.L.)
[2] Shenzhen Key Laboratory of Internet Information Collaboration, Shenzhen 518055, China
[3] Department of Information Systems, City University of Hong Kong, Hong Kong 999077, China;
   raylau@cityu.edu.hk
[4] School of Computer Science and Engineering, Nanyang Technological University,
   Singapore 628798, Singapore; Shijian.Lu@ntu.edu.sg
[5] School of Information Engineering, East China Jiaotong University, Nanchang 330000, China;
   2854@ecjtu.edu.cn
\* Correspondence: zhangxiaofeng@hit.edu.cn (X.Z.); yeyunming@hit.edu.cn (Y.Y.)

**Abstract:** Accurate hyperspectral image classification has been an important yet challenging task for years. With the recent success of deep learning in various tasks, 2-dimensional (2D)/3-dimensional (3D) convolutional neural networks (CNNs) have been exploited to capture spectral or spatial information in hyperspectral images. On the other hand, few approaches make use of both spectral and spatial information simultaneously, which is critical to accurate hyperspectral image classification. This paper presents a novel Synergistic Convolutional Neural Network (SyCNN) for accurate hyperspectral image classification. The SyCNN consists of a hybrid module that combines 2D and 3D CNNs in feature learning and a data interaction module that fuses spectral and spatial hyperspectral information. Additionally, it introduces a 3D attention mechanism before the fully-connected layer which helps filter out interfering features and information effectively. Extensive experiments over three public benchmarking datasets show that our proposed SyCNNs clearly outperform state-of-the-art techniques that use 2D/3D CNNs.

**Keywords:** convolutional neural network; 3D CNN; hyperspectral image classification

## 1. Introduction

With the rapid development of optics and photonics, hyperspctral sensors have been installed in many satellites. Hyperspectral image classification is a fundamental and yet challenging task whose purpose is to label each pixel contained in a hyperspectral image. Based on the rich spatial–spectral information preserved in hyperspectral images, it enables us to distinguish different objects of interest in the scene. They have been widely used in a variety of fields such as precise agriculture, environmental surveillance, and astronomy [1]. For example, Brown et al. [2] proposed a linear mixture model by combining various absorption band methods on CRISM to determine the mineralogy of the surface on Mars.

Hyperspectral images classification has attracted more and more research attention. Conventional image classification techniques such as support vector machine (SVM) [3] and K-nearest neighbor (KNN) classifier have achieved reasonable performance for this task as they take into account rich spectral information [4] captured in the hyperspectral images. Wang et al. [5] classified the

hyperspectral images by using the Locality Adaptive Discriminant Analysis (LADA) algorithm to reduce the dimensionality of hyperspectral images. There also exist several other approaches to cope with this issue. For example, Wang et al. [6] proposed a dimensionality reduction method for hyperspectral image classification by utilizing the manifold ranking algorithm to perform band selection. Furthermore, Yuan et al. [7] proposed a novel dual clustering-based band selection approach for hyperspectral image classification. While these methods have demonstrated superior classification performance, they are not effective to classify hyperspectral images under complex scenarios.

Recently, with the big success of deep learning, the convolutional neural networks (CNNs) [8–11] based approaches have achieved excellent performance for various image analysis related tasks, e.g., image classification and object recognition. To classify hyperspectral images, both the spectral and spatial perspectives should be taken into account. Intuitively, hyperspectral image consists of hundreds of "images", each of which represents a narrow wavelength band (visible or none-visible) of the electromagnetic spectrum, also known as spectral perspective. Meanwhile, the spatial perspective refers to the 2D spatial information about the objects contained in the hyperspectral images. Thus, hyperspectral images are usually represented as the 3D spectral–spatial data. Consequently, several approaches have been proposed in the literature. However, existing CNN-based approaches [12,13], which focus on either spatial or spectral features alone, inevitably overlook the interweaving relationships between the spatial and the spectral perspectives of objects captured in the hyperspectral images. Essentially, the interweaving information could be leveraged to further improve classification performance due to the following reasons. First, the interweaving relationships can further enrich the feature space for the later classification task. Second, if spectral–spatial information could be used simultaneously, the classifier could be co-supervised by different perspectives of the labeled data. As a result, a relatively good CNN-based classifier can be trained by using a small number of labeled 3D spectral–spatial images only.

To simultaneously model spectral–spatial information, some pioneer attempts have been made along this line [14–23]. These approaches perform stacked convolution operations over spatial and spectral feature space in a layer by layer manner, called 3D CNN models. Obviously, the advantage of this kind of 3D CNN model lies in the generated rich feature maps. However, the main disadvantages of these approaches are threefold. First, it is difficult to generate a deeper 3D CNN model. The reason is that the solution space exponentially increases with the increasing number of 3D convolution operations, which limits the depth of the model and the interpretation ability of the model. Second, the memory cost is too high if a large number of 3D convolution operations are invoked. Third, more training examples are needed to train a deeper 3D CNN model which is not practical as the public hyperspectral image datasets are rather small. To cope with aforementioned challenges, this paper is proposed to design a novel 3D CNN model which only needs few 3D convolution operations but can generate richer feature maps.

In this paper, we propose the Synergistic 2D/3D Convolutional Neural Network (SyCNN) to classify hyperspectral images. In the proposed SyCNN, the 2D CNN component and the 3D CNN component are mixed together. Different from the conventional 3D CNNs that stack up 3D convolution layer by layer, the proposed SyCNN, as shown in Figure 1, integrates 3D CNNs with 2D CNNs to learn the salient features. Then, a data interaction module is proposed which fuses the 2D features and 3D features together. Last but not least, a 3D attention mechanism is designed to drop out the less important features. Experimental results on three benchmark datasets have demonstrated that the proposed SyCNNs outperform a number of state-of-the-art hyperspectral image classifiers. Our major research contributions are summarized as follows.

1.  First, we design a basic SyCNN model (SyCNN-S) that comprises 2D CNNs and 3D CNNs to extract rich feature maps. To classify the hyperspectral images, 2D CNNs and 3D CNNs are separately trained in the SyCNN-S model before the final-connected layer. Then, the 2D output features and 3D output features are concatenated and sent to the final-connected layer to label the pixel by the softmax function.

2.　Second, we design an end-to-end hyperspectral image classification framework—deep SyCNN (SyCNN-D) by integrating data interaction module into the mixture model to generate richer feature maps. At each round of the model fusion process, the 2D CNNs and 3D CNNs are equally fused together to generate deeper and more sophisticated feature maps, and then data interaction is invoked to support the cross-domain transfer.

3.　Third, we propose a more sophisticated SyCNN model, namely SyCNN-ATT, by incorporating the 2D and 3D attention mechanisms into the SyCNN model to drop out the less relevant features.

4.　Finally, a number of extensive experiments have been conducted based on three different datasets, and the experimental results have demonstrated that the proposed SyCNN model is superior to other state-of-the-art hyperspectral image classifiers.

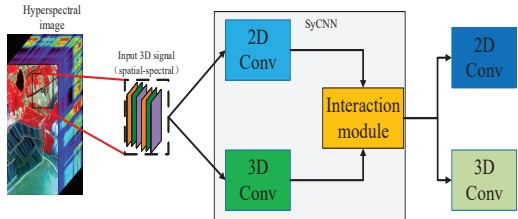

**Figure 1.** The Synergistic Convolutional Neural Network (SyCNN) framework which integrates 2D CNN, 3D CNN and the data interaction module for hyper-spectral image classification.

The rest of the paper is organized as follows. Section 2 briefly reviews the related work. In Section 3, we first carefully design the architecture of the proposed SyCNN as well as the data interaction module, and then we propose the deep SyCNNs for hyperspectral image classification task. Experimental results are reported in Section 4 and we conclude the paper in Section 5.

## 2. Related Work

Hyperspectral image classification has long been studied in the literature. However, most existing works are based on the conventional techniques. In this section, we only review the latest deep learning-based approaches and these approaches can be classified into two categories, i.e., Traditional Classification Methods and Deep Learning Models.

### 2.1. Traditional Classification Methods

A number of hyperspectral images classification methods had been proposed recently. For example, Bandos et al. [24] proposed a linear discriminant method to classify hyperspectral images. However, it cannot perform feature selection on low spectral resolution images. To automatically separate overlapping absorption bands, Bandos et al. [25] further developed a noise-insensitive method. There also exist some methods such as quadratic discriminant analysis and logarithmic discriminant analysis, which are developed to address the nonlinearity. While these methods have achieved good results, they suffer from the Huges phenomenon. For example, Wang et al. [5] proposed a novel dimensionality reduction method, named LADA to classify the hyperspectral images. Wang et al. [6] also proposed a manifold ranking based salient selection method for hyperspectral images classification. This method first finds some representative bands by employing an evolution algorithm, and then selects salient bands by using a manifold ranking strategy.

Many researchers developed kernel-based methods to improve the model performance. The kernel-based methods firstly projected samples into a high dimensional space, and then the projected samples of different classes are linearly separable. Camps et al. [26] introduced the SVM as a kernel trick in the hyperspectral image classification. Rakotomamonjy et al. [27] proposed a multiple kernel learning (MKL) method to classify hyperspectral images, which can learn a kernel and a classification predictor at the same time. However, the kernel-based methods cloud not explicitly exploit a spatial context. The composite kernel (CK) method was proposed to address

this problem. In [28], the CK method was generalized by utilizing extended multi-attribute profiles (EMAP). A generalized composite kernel (GCK) was proposed to exploit both extended multi-attribute profiles and raw hyperspectral features in [29].

### *2.2. Deep Learning Models*

Recently, deep learning models have achieved a significant performance on image classification task. Deep learning based methods are introduced into hyperspectral images classification problem to learn hierarchical representations from the hyperspectral remote sensing images.

### 2.2.1. 2D CNN-Based Approaches

LeCun firstly deigned the CNN [30,31] model on 1996. Due to the back propagation (BP) mechanism, the proposed CNN model can be trained easily and achieves a very good performance in handwritten digit recognition. A lot of deep learning based methods are developed in many fields. For example, Glorot et al. [32] improved the CNNs by introducing the Rectified Linear Units (ReLU) as the activation function. It can alleviate the vanishing gradient problem and the ineffective exploration problem caused by the BP mechanism. Krizhevsky et al. [8] designed the AlexNet network with ReLU activation. However, the deeper of the deep learning network will lead to the overfitting problem. To address this phenomenon, Srivastava et al. [33] proposed the dropout mechanism after the convolutional operation to reduce the noisy samples. At the same time, Szegedy et al. [10] proposed the GoogLeNet model to classify the images, which is a deep CNN model with each layer comprising multi-scale convolutional operations. He et al. [11] introduced the residual mechanism in the network, and proposed a deep residual CNN model, namely ResNet.

There also have a lot of works for hyperspectral image classification based on CNN. To extract the spectral–spatial information contained in hyperspectral images, a 2D CNN-based approach was proposed in [12], where 2D CNN was utilized to explore the band selection results generated by the adapted adaBootst-based SVM classifier. Based on the band selection results, several methods were proposed to fuse certain 2D CNN networks for hyperspectral images classification. For example, Liu et al. [34] utilized a deep belief netowkrs (DBNs) to extract deep spectral features. Ran et al. [35] proposed a deep band sensitive convolutional network to classify the hyperspectral images and the proposed network has two sub-components. One component is to extract the spectral features and another one is to aggregate there features from different bands. In [36], the author proposed an attention-based convolutional neural network to perform band-selection for hyperspectral images. Furthermore, Liu et al. [13] proposed a semi-supervised 2D CNN model including the encoder, the corrupted encoder and the decoder component. The classification model is trained to minimize the reconstruction error between the input images and the mixture of both labeled and unlabeled images. One of the state-of-the-art approach is proposed in [14]. In this paper, a deep 2D CNN model is proposed to directly label each pixel in the hyperspectral images. Alternatively, the two-streamlike 2D CNN approaches [37,38] were proposed with each stream of the proposed network was a pre-trained 2D CNN model. While these approaches could achieve comparably good model performance, a large training dataset was a prerequisite which is unaffordable in most real-world applications.

### 2.2.2. Three-Dimensional CNN-Based Approaches

In [39], the 3D CNN approach was first proposed to learn discriminative features to recognize object actions from the spatial-temporal data. Later, Chen et al. [17] proposed a deep 3D CNN network which stacked up several 3D convolutional layers to extract spectral–spatial feature maps for classification. Similarly, a deep fully convolutional network (FCN) with a focus on 3D data was proposed by Lee et al. [16]. Different from [16], Li et al. [18] proposed a 3D CNN network which stacked up 3D convolutional layers without the pooling layer. The proposed model can well capture the changes of local signals contained in the spectral–spatial data. Hamida et al. [15] designed a deep 3D CNN network for hyperspectral image classification, where the pooling layers were replaced

by the spectral–spatial 3D convolutional layers. Furthermore, there were some hybrid models that combined 2D CNNs with 3D CNNs. Obviously, the 3D CNN-based approaches have more parameters than the 2D CNN-based approaches. Accordingly, both the model complexity and the memory consumption of the 3D CNN-based approaches are much higher than that of the 2D CNN-based approach. As a result, Sun et al. [40] tried to replace the 3D convolutional layer with a mixture of a 2D spatial convolutional layer and a 1D temporal convolutional layer which could largely alleviate the aforementioned problem. Unfortunately, this approach extracted the spatial features and spectral features separately and thus ignored the interweaving relationships between the spatial and the spectral perspectives of the hyperspectral images.

Apparently, these 3D CNN-based approaches cannot be directly adapted to model such interweaving relationships due to the following reason. Most existing 3D CNN-based approaches have several stacked 3D convolutional layers and thus is hard to directly optimize the estimation loss through such nonlinear structure. To take the advantages of both the 2D and 3D CNN-based approaches, this paper proposed to fuse the features extracted from 2D and 3D convolution operations in an iterative manner and synergistically trained a hybrid 2D/3D CNN model with relatively few training examples.

## 3. The Proposed SyCNN and Deep SyCNN Network

In this section, we first introduce the 3D convolution operation. Then, we introduce the proposed SyCNN with more technical details. Finally, we propose a deep SyCNN network for hyperspectral image classification task.

### 3.1. 3D Convolution Operation

Generally speaking, a 3D spectral–spatial hyperspectral image can be represented by a tensor and its size is $S \times H \times W \times C$, where $S$ denotes the spectral domain (comprising a range of wavelengths), $H$ and $W$ are the height and width in the spatial domain, and $C$ represents the number of channels. Different from the 2D convolution operation, the kernels of a 3D convolution operation are formulated as a 4D tensor $\psi \in \Re^{n_k \times s_k \times h_k \times w_k}$, where $s_k, h_k, w_k$ denote the kernel size in $S, H$, and $W$, respectively, and $n_k$ is the number of kernels. The 3D convolution operation is illustrated in Figure 2. In Figure 2, it is easy to observe that the 3D convolution operation takes the 3D spectral–spatial features $I = I_{s,h,w}$ as the inputs and produces the 3D feature maps $O = o_{s,h,w}$ by executing the convolution operation along both the spatial and spectral dimensions of the inputs. Accordingly, the 3D convolution operation can be formulated as follows:

$$
\begin{aligned}
O &= \psi \otimes I, \\
o_{s_0,h_0,w_0} &= [q^1_{s_0,h_0,w_0},\ q^2_{s_0,h_0,w_0}, \cdots,\ q^{n_k}_{s_0,h_0,w_0}]^T, \\
q^n_{t_0,h_0,w_0} &= \Sigma_{t,w,h} \psi_{n,t,h,w} \cdot I^{t_0 h_0 w_0}_{s,w,h},
\end{aligned}
\tag{1}
$$

where $I^{s_0 h_0 w_0}_{s,w,h}$ denotes that the kernel starts from the position $(s_0, h_0, w_0)$ of the input image $I$ and ends at the position $(s, w, h)$ of the input $I$. The $q^n_{s_0,h_0,w_0}$ is the value at the cell of $(s_0, h_0, w_0)$ on the $n$-th output feature map by the $n$-th kernel.

Data Interaction

The proposed data interaction module for generating 2D and 3D feature maps is illustrated in Figure 3. This module consists of the cross-domain transfer operation and the local cross-domain connection operation. Let $O_2$ denote the 2D output feature maps, and $O_3$ denote the 3D output feature map. Moreover, $t$ is the time slot, then we have:

$$O_2^t = \phi(I_2^t) + I_2^t,$$
$$O_3^t = \psi(I_3^t) + I_3^t,$$

$$(2)$$

where $I_2^t \in \Re^{h \times w}$ and $I_3^t \in \Re^{s \times h \times w}$ are the 2D input features and the 3D input features at time $t$, respectively. $\phi$ denotes the 2D convolution and $\psi$ is the 3D convolution. As shown in Figure 3, the 2D features and 3D features are interacted by using the cross-domain transfer module and the local cross-domain connection module.

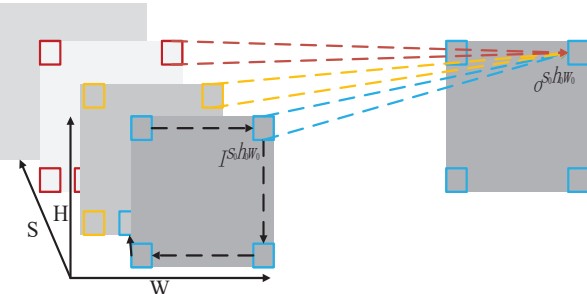

**Figure 2.** Illustration of a 3D convolution operation. The convolution kernels slide along both the spatial and spectral dimensions of the input 3D images and generate the 3D spectral–spatial features.

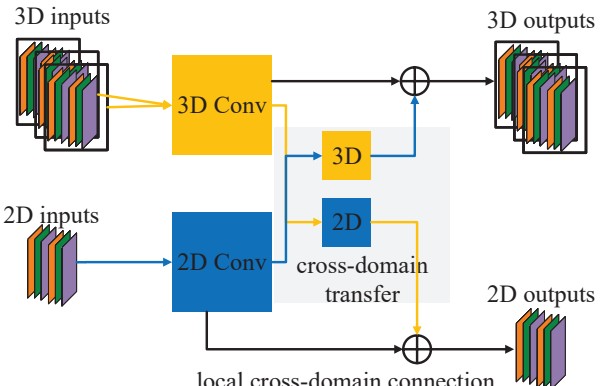

**Figure 3.** SyCNN with data interaction module that consists of the local cross-domain connection and cross-domain transfer. 2D spectral–spatial fusion is achieved by transferring the 3D inputs to 2D outputs and concatenating these outputs with the 2D inputs. On the other hand, the 3D spectral–spatial fusion is obtained by transferring the 2D inputs to 3D outputs and concatenating these 3D outputs with the 3D inputs.

We first transfer the 3D features obtained by the 3D convolution to the new 2D features $\acute{O}_2^t$, and then transfer the 2D features obtained by the 2D convolution to the new 3D features $\acute{O}_3^t$, and we have:

$$\acute{O}_2^t = \tau(O_3^t),$$
$$\acute{O}_3^t = \beta(O_2^t),$$

$$(3)$$

where $\beta$ denotes the transfer operation from 2D features to 3D features, and $\tau$ denotes the transfer operation from 3D features to 2D features. The transferred features are then concatenated with the corresponding 2D or 3D features to form the final output features, written as:

$$O_2^t = \acute{O}_2^t + O_2^t,$$
$$O_3^t = \acute{O}_3^t + O_3^t.$$

$$(4)$$

It is easy to observe that the final output 2D features $O_2^t$ contain not only the 2D spatial feature information but also the 3D spectral feature information. Similarly, the output 3D features $O_3^t$ also

contain spatial and spectral feature information. The new 2D features and 3D features will be fed into the next 2D convolutional and 3D convolutional operations as the input, respectively. By doing so, we already generate more training samples to train the 3D CNNs, and the feature maps of 2D CNNs are also spectral–spatial information. Thus the data interaction module not only reduces the complexity of the proposed SyCNN in learning 3D convolution kernels, but also generates rich feature maps to train the SyCNN. Due to its prowerful learning and extraction abilities, the proposed SyCNN can greatly alleviate the overfitting problem.

### 3.2. Deep SyCNN Network

To perform hyperspectral image classification, we design a hybrid deep model by stacking 2D convolution and 3D convolution which is synergistically trained. As shown in Figure 4a, the proposed simple mixed model consists of three 2D/3D convolutional operations, which directly connect to the fully-connected layer to classify unlabeled images. We design a simple yet efficient deep SyCNN network by stacking the SyCNNs together. The proposed SyCNN model is an end-to-end network and takes the hyperspectral images as input. The proposed deep SyCNN network consists of three SyCNNs which only involves 3D convolutions, and the model is illustrated in Figure 4b. To further improve the model performance, we introduce the attention mechanism with a focus on generating salient features to facilitate hyperspectral image classification task. The proposed deep SyCNN-attention model is plotted in Figure 4c. Furthermore, it is a none-trivial task to pre-process the input data. Note that the BN-inception [41] and the ReLU function [32] are applied after each convolutional block in the proposed model, which is used to address the overfitting problem caused by the sparse training data and limited training samples in hyperspectral images. For simplicity reason, these BN and ReLu layers are not plotted in the figure. In order to allow the input images of any length, we use a global pooling layer as the last layer of the network.

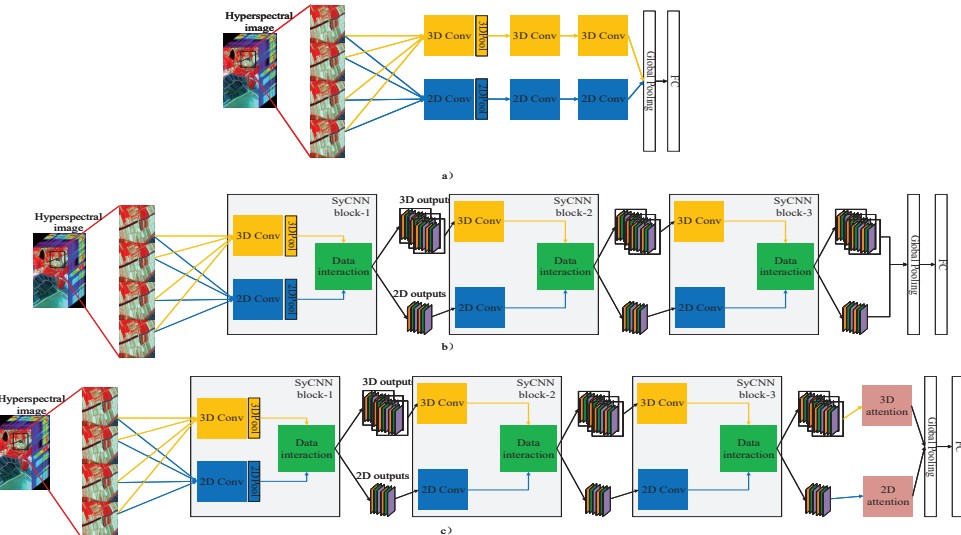

**Figure 4.** Illustration of the proposed models: (**a**) The deep simple SyCNN model, (**b**) the deep SyCNN model, (**c**) the deep SyCNN with attention module. Yellow blocks and blue blocks refer to 3D convolution and 2D convolution. Green blocks refer to data interaction module.

The difference between the proposed approach and the state-of-the-art 3D CNN models [14,19] is that the deep SyCNN requires fewer 3D convolution operations for the spectral–spatial fusion stage, and yet it can generate deeper and richer feature maps. Moreover, different from the conventional 3D CNN based models, the deep SyCNN approach can take full advantage of 2D CNN approaches, and yet it can be trained using a much smaller image data set.

## 4. Experimental Results

To evaluate the performance of the proposed deep SyCNN, rigorous experiments are performed on three benchmark hyperspectral image datasets. We implemented the proposed approaches as well as several state-of-the-art approaches on these datasets. The experimental settings as well as the evaluation criteria are given in the following subsections. The proposed approaches are called deep simple SyCNN network (SyCNN-S), deep SyCNN network with data interaction module (SyCNN-D), and deep SyCNN-attention network (SyCNN-ATT), respectively.

### 4.1. Experimental Settings

In these experiments, three widely adopted benchmark hyperspectral image datasets, i.e., Indian Pines Scene, Botswana Scene, and Kennedy Space Center datasets, are chosen for model performance evaluation.

The Indian Pines Scene dataset: This dataset is collected by the AVIRIS sensors located at north-western of India, U.S., on 1992. It contains 145 × 145 pixels in spatial dimension in the image, and 200 pixels in spectral dimension. Due to the presence of 20 noisy bands, we only used 200 hyperspectral bands in this experiment. Specifically, it removes the bands covering the regions of water absorption, i.e., [104–108], [150–163], 220. The ground truth available includes 16 classes. The false-color image and the corresponding ground reference map are shown in Figure 5. The numbers of the training and test samples per class are listed in Table 1.

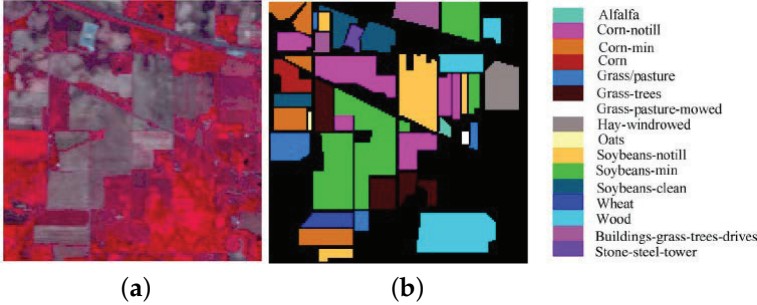

**(a)** **(b)**

**Figure 5.** Indian Pines dataset. (**a**) Three-band false-color composite. (**b**) Ground-truth map.

**Table 1.** Sample size for Indian Pines Scene.

| NO. | Class Name | Training Samples | Testing Samples |
|---|---|---|---|
| 1 | Alfalfa | 32 | 14 |
| 2 | Corn-notill | 1003 | 424 |
| 3 | Corn-mintill | 585 | 245 |
| 4 | Corn | 168 | 69 |
| 5 | Grass-pasture | 340 | 143 |
| 6 | Grass-trees | 512 | 216 |
| 7 | Grass-pasture-mowed | 20 | 8 |
| 8 | Hay-windrowed | 335 | 143 |
| 9 | Oats | 14 | 6 |
| 10 | Soybean-notill | 683 | 289 |
| 11 | Soybean-mintill | 1721 | 733 |
| 12 | Soybean-clean | 417 | 174 |
| 13 | Wheat | 144 | 61 |
| 14 | Woods | 888 | 374 |
| 15 | Buildings-Grass-Trees-Drives | 272 | 113 |
| 16 | Stone-Steel-Towers | 65 | 28 |
| | Total | 7200 | 3040 |

The Botswana Scene dataset: This dataset is collected by the Hyperion sensors on the NASA EO-1 satellite over the Okavango Delta on 31 May 2001. It has 1476 × 1476 pixels in the spatial dimension and

145 corrected bands in spectral dimension. Following with the Indian Pines Scene dataset, we removed the noisy bands to produce an experimental data set containing 145 bands. This image dataset contains 14 categories. The false-color image and the corresponding ground reference map are shown in Figure 6. The numbers of the training and test samples per class are listed in Table 2.

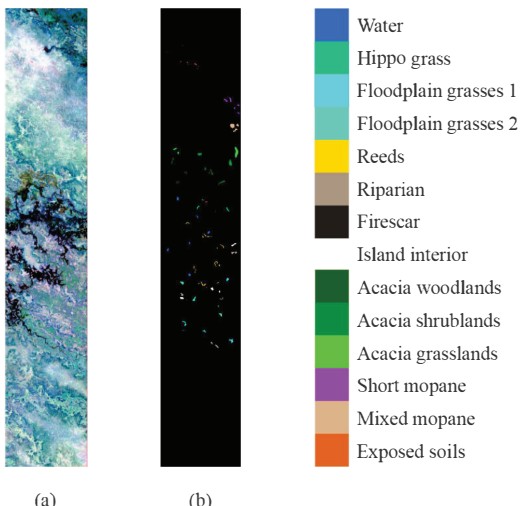

(a)       (b)

**Figure 6.** Botswana Scene dataset. (**a**) Three-band false-color composite. (**b**) Ground-truth map.

**Table 2.** Sample size for Botswana Scene.

| NO. | Class Name | Training Samples | Testing Samples |
|-----|------------|------------------|-----------------|
| 1 | Water | 189 | 81 |
| 2 | Hippo grass | 71 | 30 |
| 3 | Floodplain grasses 1 | 176 | 75 |
| 4 | Floodplain grasses 2 | 151 | 64 |
| 5 | Reeds | 188 | 81 |
| 6 | Riparian | 188 | 81 |
| 7 | Fire scar | 183 | 76 |
| 8 | Island interior | 143 | 60 |
| 9 | Acacia woodlands | 220 | 94 |
| 10 | Acacia shrunblands | 174 | 74 |
| 11 | Acacia grasslands | 214 | 91 |
| 12 | Short mopane | 127 | 54 |
| 13 | Mixed mopane | 189 | 79 |
| 14 | Exposed soils | 67 | 28 |
| | Total | 2280 | 968 |

The Kennedy Space Center (KSC) dataset: This dataset is captured by using the AVIRIS sensor over Florida on 23 March 1996. It has $512 \times 614 \times 172$ pixels in spectral–spatial dimension. Dut to remove 48 noisy bands, it obtains 172 spectral bands. The image dataset contains 13 labeled classes. The false-color image and the corresponding ground reference map are shown in Figure 7. The numbers of the training and test samples per class are listed in Table 3.

In the experiments, these datasets are randomly partitioned into training and testing datasets at the ratio of 70% to 30%. The proposed approaches are implemented on a NVIDIA GTX1080ti GPU machine using Pytorch [42]. The convolutional operation with the same color share the same parameters of each proposed model. We adopt the Adam gradient descent optimizer with an initial learning rate of $1e^{-4}$ to train our models. The mini-batch size is set to 100. The drop out ratio and weight decay rate are respectively set to 0.5 and $5e^{-4}$.

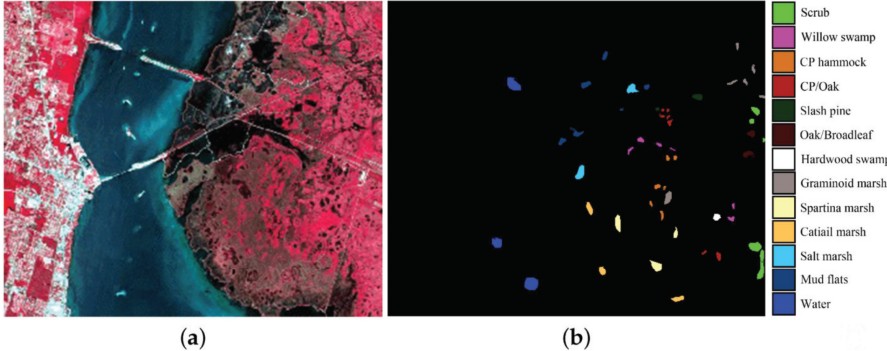

**Figure 7.** Kennedy Space Center dataset. (**a**) Three-band false-color composite. (**b**) Ground-truth map.

**Table 3.** Samples size for Kennedy Space Center scene.

| NO. | Class Name | Training Samples | Testing Samples |
|-----|------------|------------------|-----------------|
| 1 | Scrub | 533 | 228 |
| 2 | Willow swamp | 170 | 73 |
| 3 | Cabbage palm hammock | 179 | 77 |
| 4 | Cabbage palm/oak hammock | 176 | 76 |
| 5 | Slash pine | 113 | 48 |
| 6 | Oak/broadleaf hammock | 160 | 69 |
| 7 | Hardwood swamp | 74 | 31 |
| 8 | Graminoid marsh | 302 | 129 |
| 9 | Spartina marsh | 364 | 156 |
| 10 | Cattail marsh | 284 | 120 |
| 11 | Salt marsh | 294 | 125 |
| 12 | Mud flats | 352 | 151 |
| 13 | Water | 649 | 278 |
| | Total | 3650 | 1561 |

### 4.2. Baseline Models and Evaluation Criteria

We compare our approaches with both conventional and the state-of-the-art approaches which are illustrated as follows.

- **SVM-Grid** A support vector machine based approach optimized by the stochastic gradient descent algorithm. This conventional approach is one of the most widely adopted algorithms for hyperspectral image classification.
- **NN** A simple neural network (NN) model with 4 fully connected layers with dropout, and this approach is considered as the baseline deep learning model.
- **Sharma** A 2D CNN model consists of three 2D convolutional operations with band selection [12], originally proposed for hyperspectral face recognition. It is a dimension reduction method, where 1D convolutional operation is introduced to lead to a drastic reduction in the number of parameters used.
- **Liu** is a semi-supervised 2D CNN model [13] for hyperspectral image classification, and it includes one convolutional operation, one clean encoder, one corrupted encoder, and one decoder.
- **Hamida** is a 3D deep learning approach for hyperspectral image classification consisting of four 3D convolutions [15].
- **Lee** is a fully convolutional network (FCN) [43] which does not use any subsampling layer with arbitrary sizes. It contains two 3D convolutions and eight 2D convolutions [16].
- **Chen** is a deep 3D CNN model consisting of three 3D convolutions [17].

- **Li** is a 3D CNN model having two 3D convolutions and a fully-connected layer for image classification [18]. Different from other 3D CNNs, the 3D convolution in this work has a fixed spatial size and only slightly changes its spectral depth.
- **He** is a multi-scale 3D deep learning network [19] consisting of two 3D convolutions and two 3D convolution-blocks, and each block contains four 3D convolutions.

### 4.3. Evaluation Criteria

Three widely adopted evaluation criteria are chosen in the experiments to evaluate the model performance which are overall accuracy (OA), the F1 score and Kappa coefficient (K). The OA is the mean accuracy of each category. The F1 score is also called balanced F score which is harmonic average value of precision and recall value of each category. The Kappa coefficient (K) is an index value measuring whether the classification results are consistent with the underlying true classes or not. For hyperspectral image classification, we mainly check whether such inconsistency is caused by "accidental" factors or "inevitable" factors. These evaluation criteria are calculated as follows.

$$
\begin{aligned}
Precison &= \frac{TP}{TP + FP}, \\
Recall &= \frac{TP}{TP + FN}, \\
Accuracy &= \frac{TP + TN}{TP + TN + FN + FP}, \\
OA &= \left(\frac{1}{n} \sum_k Accuracy_k\right), \\
F1 &= \left(\frac{1}{n} \sum_k \frac{2 \cdot Precison_k \cdot Recall_k}{Precison_k + Recall_k}\right)^2, \\
K &= \frac{N \sum_{i=1}^{n} x_{ii} - \sum_{i=1}^{n}(x_{i+} \times x_{+i})}{N^2 - \sum_{i=1}^{n}(x_{i+} \times x_{+i})}.
\end{aligned}
\tag{5}
$$

where $TP$ denotes the true positive value, $FP$ denotes the false positive value, and $FN$ denotes the false negative value. $n$ is the number of categories, and $N$ is the total number of data samples. $x_{ii}$ denotes the number of categories on the diagonal of the confusion matrix, $x_{i+}$ is the total number of the $i$-th row, and $x_{+i}$ denotes the total number of the $i$-th column.

### 4.4. Experimental Results

4.4.1. Experimental Results on the Indian Pines Scene Dataset

The comparative model performance based on the Indian Pines Scene dataset with respect to OA, F1 and K are depicted in Table 4. From this table, it is obvious that the model performance of our proposed approaches is superior to both baseline methods and the state-of-the-art approaches. Among all proposed approaches, the SyCNN is the best one with respect to all evaluation criteria and this is consistent with our expectation. Moreover, the proposed deep SyCNN-attention model can achieve a comparably good model performance to the proposed SyCNN. Among all compared methods, the Sharma achieves the best classification results. It is also well noticed that the performance of 2D CNN based approaches is better than that of 3D CNN based approaches. Apparently, the reason is that the number of training examples is rather limited which only favors the less complicate models such as the Sharma.

To further evaluate the model performance, the experimental results of all approaches are plotted in Figure 8. It can be seen from Figure 8 that the proposed SyCNN model performs the best result among all methods. The possible reasons are as follows: (i) It is a synergistic trained model, which consists of 2D convolutions and 3D convolutions to generate deeper and richer features; (ii) it makes full use of the 2D and 3D features by the data interaction module. Since the parameters of the

deep 3D CNN models need more training samples, we find that the Sharma and Liu methods produce the second better results than the methods based on 3D CNNs, such as Hamida, Lee, Li, Chen, and He. It's observed that the worst model is NN, which is a baseline deep learning model for hyperspectral image classification. The reasons are that the NN model only has one 2D convolution but 4 fully connected layers for prediction task and the 2D convolution is not sufficient enough to generate the deep and rich features.

**Table 4.** Evaluation results on the Indian Pines Scene dataset.

| Class | Methods | | | | | | | | | | | |
|---|---|---|---|---|---|---|---|---|---|---|---|---|
| # | SVM-Grid | NN | Sharma | Liu | Hamida | Lee | Li | Chen | He | SyCNN-S | SyCNN-D | SyCNN-ATT |
| 1 | 88.00 | 92.30 | 100 | 66.70 | 100 | 63.60 | 100 | 96.30 | 100 | 100 | 100 | 100 |
| 2 | 85.50 | 84.20 | 99.40 | 88.50 | 84.90 | 82.40 | 96.60 | 95.90 | 89.00 | 100 | 99.60 | 99.20 |
| 3 | 80.80 | 80.30 | 93.60 | 86.10 | 79.70 | 80.40 | 90.60 | 91.10 | 86.00 | 91.30 | 91.70 | 96.10 |
| 4 | 79.20 | 71.90 | 100 | 83.60 | 93.60 | 82.80 | 95.80 | 97.20 | 95.00 | 97.10 | 100 | 99.30 |
| 5 | 95.20 | 93.70 | 94.60 | 91.60 | 93.80 | 93.70 | 94.50 | 92.60 | 95.70 | 96.10 | 95.30 | 97.20 |
| 6 | 96.70 | 96.10 | 100 | 98.40 | 99.10 | 97.50 | 99.80 | 99.10 | 100 | 100 | 99.50 | 100 |
| 7 | 80.00 | 93.30 | 100 | 40.00 | 76.90 | 82.40 | 100 | 93.30 | 100 | 100 | 100 | 100 |
| 8 | 99.00 | 99.30 | 100 | 96.30 | 99.00 | 97.30 | 99.70 | 99.70 | 100 | 100 | 100 | 100 |
| 9 | 66.70 | 100 | 100 | 0 | 90.90 | 28.30 | 100 | 100 | 50.00 | 100 | 100 | 100 |
| 10 | 80.60 | 82.40 | 97.00 | 89.30 | 76.20 | 84.20 | 96.10 | 90.50 | 91.90 | 97.70 | 97.00 | 98.30 |
| 11 | 84.90 | 87.90 | 97.50 | 91.50 | 87.40 | 89.80 | 96.20 | 94.70 | 92.60 | 98.10 | 98.00 | 98.70 |
| 12 | 89.30 | 83.10 | 97.20 | 89.70 | 87.70 | 79.70 | 96.20 | 96.90 | 93.80 | 98.30 | 98.90 | 98.60 |
| 13 | 100 | 100 | 100 | 100 | 100 | 96.10 | 100 | 100 | 100 | 100 | 100 | 100 |
| 14 | 95.80 | 93.50 | 99.70 | 98.40 | 98.90 | 94.20 | 99.70 | 98.90 | 99.10 | 100 | 100 | 100 |
| 15 | 76.70 | 69.10 | 84.20 | 77.20 | 76.30 | 72.00 | 82.20 | 83.70 | 85.80 | 86.80 | 91.10 | 87.40 |
| 16 | 100 | 98.20 | 96.40 | 100 | 98.20 | 100 | 100 | 98.20 | 96.40 | 100 | 98.20 | 100 |
| OA | 87.93 | 87.57 | 95.64 | 89.56 | 86.99 | 87.87 | 94.22 | 93.20 | 91.87 | 95.90 | 96.13 | **97.31** |
| F1 | 87.40 | 89.07 | 97.48 | 81.08 | 90.16 | 83.42 | 96.71 | 95.51 | 92.21 | 97.84 | 98.08 | **98.43** |
| K | 86.20 | 85.80 | 95.10 | 88.10 | 85.20 | 86.10 | 93.40 | 92.30 | 90.80 | 95.30 | 95.60 | **96.90** |

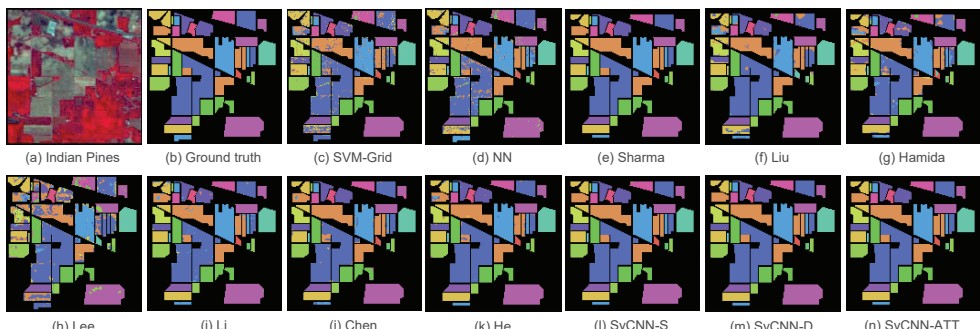

(a) Indian Pines   (b) Ground truth   (c) SVM-Grid   (d) NN   (e) Sharma   (f) Liu   (g) Hamida

(h) Lee   (i) Li   (j) Chen   (k) He   (l) SyCNN-S   (m) SyCNN-D   (n) SyCNN-ATT

**Figure 8.** Visualization of the experimental results based on Indian Pines: (**a**) Indian Pines, (**b**) Ground truth, (**c**) SVM-Grid, (**d**) NN, (**e**) Sharma, (**f**) Liu, (**g**) Hamida, (**h**) Lee, (**i**) Li, (**j**) Chen, (**k**) He, (**l**) SyCNN-S, (**m**) SyCNN-D, (**n**) SyCNN-ATT. It is observed that the outputs produced by our proposed models are quite close to the Ground truth.

According to the Table 4, the Sharma approach can achieve the best model performance among all compared models. However, the model performance of all proposed approaches is better than that of the Sharma. Among all proposed approaches, the results of the evaluation criteria, i.e., OA, F1 and K, of the proposed SyCNN-S is respectively increased by 0.26%, 0.36% and 0.2% when compared with the Sharma. The SyCNN-D can further improve the model performance by 0.228%, 0.24% and 0.3% with respect to OA, F1 and K. The SyCNN-ATT is the best one among all proposed approaches. Apparently, the 3D CNN based approaches cannot achieve desired better model performance than 2D CNN based approaches which could also be found in the visualized results plotted in Figure 8.

### 4.4.2. Experimental Results Based on the Botswana Scene Dataset

Table 5 reports the evaluation results based on the Botswana Scene dataset. Botswana Scene dataset only has 3248 samples, and the labels are unbalanced. Similarly, the proposed approaches can achieve the best classification performance. Due to the data interaction module, the proposed SyCNN-D could extract the deeper and richer features from the hyperspectral images. The 2D convolutional operation could extract much more spatial information by treating the whole spectral information as channels, but it loses the relationship between different spectral bands. While the 3D convolutional operation cloud extract much more spatial–spectral fusion information by operating on spectral information, it could only combine a few spectral information with a fixed size and need more training samples for the next operation. In the data interaction module, the 2D extracted features could be projected into 3D features and the projected 3D features are fused in the 3D extracted features. Thus, the new 3D features combine more spectral information and make full use of spatial information. In addition, the new 3D features are sufficient to train the next 3D convolutional operation. So as the new 2D features that have more spectral information are also sufficient to train the next 2D convolutional operation. The Sharma and Liu methods are the second best models, and the rest 3D CNN based approaches are the worst models except for the NN model.

**Table 5.** Comparative evaluation based on the Botswana Scene dataset.

| Class | Methods | | | | | | | | | | | |
|---|---|---|---|---|---|---|---|---|---|---|---|---|
| # | SVM-Grid | NN | Sharma | Liu | Hamida | Lee | Li | Chen | He | SyCNN-S | SyCNN-D | SyCNN-ATT |
| 1 | 99.40 | 100 | 99.40 | 100 | 100 | 100 | 100 | 100 | 100 | 100 | 100 | 100 |
| 2 | 98.40 | 96.80 | 100 | 96.60 | 92.90 | 98.40 | 100 | 100 | 100 | 98.40 | 100 | 100 |
| 3 | 98.60 | 92.90 | 100 | 100 | 96.60 | 91.70 | 100 | 100 | 100 | 100 | 100 | 100 |
| 4 | 97.70 | 90.30 | 100 | 100 | 92.70 | 69.30 | 99.20 | 91.40 | 97.70 | 100 | 100 | 100 |
| 5 | 92.40 | 81.50 | 98.80 | 95.20 | 91.00 | 83.70 | 94.90 | 97.70 | 93.40 | 98.80 | 100 | 100 |
| 6 | 87.40 | 74.40 | 99.40 | 92.10 | 72.60 | 76.10 | 91.00 | 92.90 | 91.30 | 100 | 100 | 100 |
| 7 | 98.70 | 99.40 | 100 | 100 | 100 | 99.40 | 100 | 89.30 | 100 | 100 | 100 | 100 |
| 8 | 100 | 97.50 | 100 | 100 | 97.60 | 76.90 | 100 | 98.10 | 100 | 100 | 100 | 100 |
| 9 | 90.20 | 85.10 | 100 | 96.10 | 64.70 | 61.70 | 100 | 93.80 | 97.40 | 100 | 100 | 100 |
| 10 | 90.30 | 83.90 | 99.30 | 97.90 | 97.40 | 81.70 | 100 | 94.40 | 100 | 100 | 100 | 100 |
| 11 | 94.40 | 93.10 | 99.50 | 97.90 | 95.50 | 95.60 | 100 | 91.50 | 100 | 100 | 100 | 100 |
| 12 | 91.40 | 95.40 | 98.10 | 99.10 | 97.10 | 90.60 | 96.20 | 96.20 | 99.10 | 95.20 | 97.10 | 99.10 |
| 13 | 93.90 | 87.70 | 100 | 100 | 99.40 | 100 | 100 | 99.40 | 100 | 100 | 100 | 100 |
| 14 | 100 | 96.70 | 100 | 100 | 90.60 | 88.50 | 100 | 92.60 | 100 | 100 | 100 | 100 |
| OA | 94.66 | 90.25 | 99.48 | 97.49 | 91.38 | 85.94 | 97.94 | 95.38 | 98.25 | 99.38 | 99.69 | **99.79** |
| F1 | 95.20 | 91.05 | 99.61 | 99.68 | 92.01 | 86.34 | 98.36 | 95.52 | 98.49 | 99.46 | 99.79 | **99.93** |
| K | 94.20 | 89.40 | 99.40 | 97.80 | 90.70 | 84.80 | 97.80 | 95.00 | 98.10 | 99.30 | 99.70 | **99.80** |

We also visualize the corresponding experimental results evaluated based on Botswana Scene dataset in Figure 9. Clearly, the SyCNN-ATT model is the best one among all the experimental methods.

### 4.4.3. Experimental Results Based on the Kennedy Space Center Dataset

Table 6 reports the evaluation results based on the Kennedy Space Center dataset. Similar observations of the proposed methods are obtained, that is, they achieve better classification performance than other baseline methods. Among all proposed methods, the SyCNN-ATT method is the best one. Again, the outstanding performance of the proposed deep SyCNN-ATT model can be easily observed based on the visualized results that are plotted in Figure 10.

**Table 6.** Classification results of the Salinas scene.

| Class | Methods | | | | | | | | | | | |
|---|---|---|---|---|---|---|---|---|---|---|---|---|
| # | SVM-Grid | NN | Sharma | Liu | Hamida | Lee | Li | Chen | He | SyCNN-S | SyCNN-D | SyCNN-ATT |
| 1 | 57.80 | 87.60 | 89.40 | 81.50 | 69.40 | 94.50 | 87.40 | 54.10 | 91.40 | 94.40 | 94.70 | 97.80 |
| 2 | 72.50 | 71.30 | 94.50 | 80.00 | 61.80 | 68.80 | 91.20 | 88.60 | 93.40 | 98.60 | 99.30 | 99.30 |
| 3 | 0 | 62.40 | 92.20 | 68.30 | 7.50 | 72.20 | 69.20 | 91.40 | 95.10 | 96.70 | 100 | 100 |
| 4 | 0 | 24.10 | 76.60 | 60.90 | 40.70 | 7.30 | 70.30 | 91.50 | 75.20 | 94.60 | 98.70 | 100 |
| 5 | 0 | 53.70 | 88.90 | 61.70 | 31.60 | 67.40 | 82.20 | 100 | 86.30 | 95.90 | 97.90 | 96.80 |
| 6 | 0 | 30.80 | 67.30 | 42.20 | 2.90 | 64.20 | 76.30 | 98.50 | 87.80 | 98.60 | 99.30 | 100 |
| 7 | 0 | 54.20 | 92.50 | 85.30 | 5.60 | 35.00 | 90.00 | 96.80 | 98.40 | 98.50 | 100 | 100 |
| 8 | 41.60 | 61.40 | 94.30 | 72.70 | 70.20 | 80.00 | 81.40 | 82.80 | 97.20 | 98.00 | 96.40 | 97.70 |
| 9 | 73.40 | 83.10 | 99.40 | 84.50 | 85.30 | 95.70 | 91.10 | 93.30 | 99.40 | 100 | 100 | 100 |
| 10 | 83.50 | 88.10 | 99.60 | 96.60 | 77.10 | 86.30 | 96.70 | 100 | 99.20 | 100 | 100 | 100 |
| 11 | 96.00 | 98.40 | 100 | 100 | 99.60 | 98.80 | 99.60 | 100 | 100 | 100 | 100 | 100 |
| 12 | 74.80 | 85.30 | 99.70 | 95.00 | 87.10 | 85.00 | 93.50 | 100 | 99.00 | 99.70 | 100 | 100 |
| 13 | 96.70 | 99.50 | 100 | 99.60 | 99.80 | 97.30 | 100 | 100 | 100 | 100 | 100 | 100 |
| OA | 67.39 | 79.79 | 93.35 | 84.27 | 76.22 | 84.14 | 88.62 | 86.83 | 94.69 | 97.44 | 97.76 | **98.92** |
| F1 | 45.89 | 67.37 | 91.88 | 79.10 | 62.82 | 73.27 | 86.84 | 92.08 | 94.03 | 98.46 | 98.95 | **99.35** |
| K | 62.80 | 77.40 | 92.60 | 82.50 | 73.10 | 82.30 | 87.40 | 85.50 | 94.10 | 97.20 | 97.50 | **98.80** |

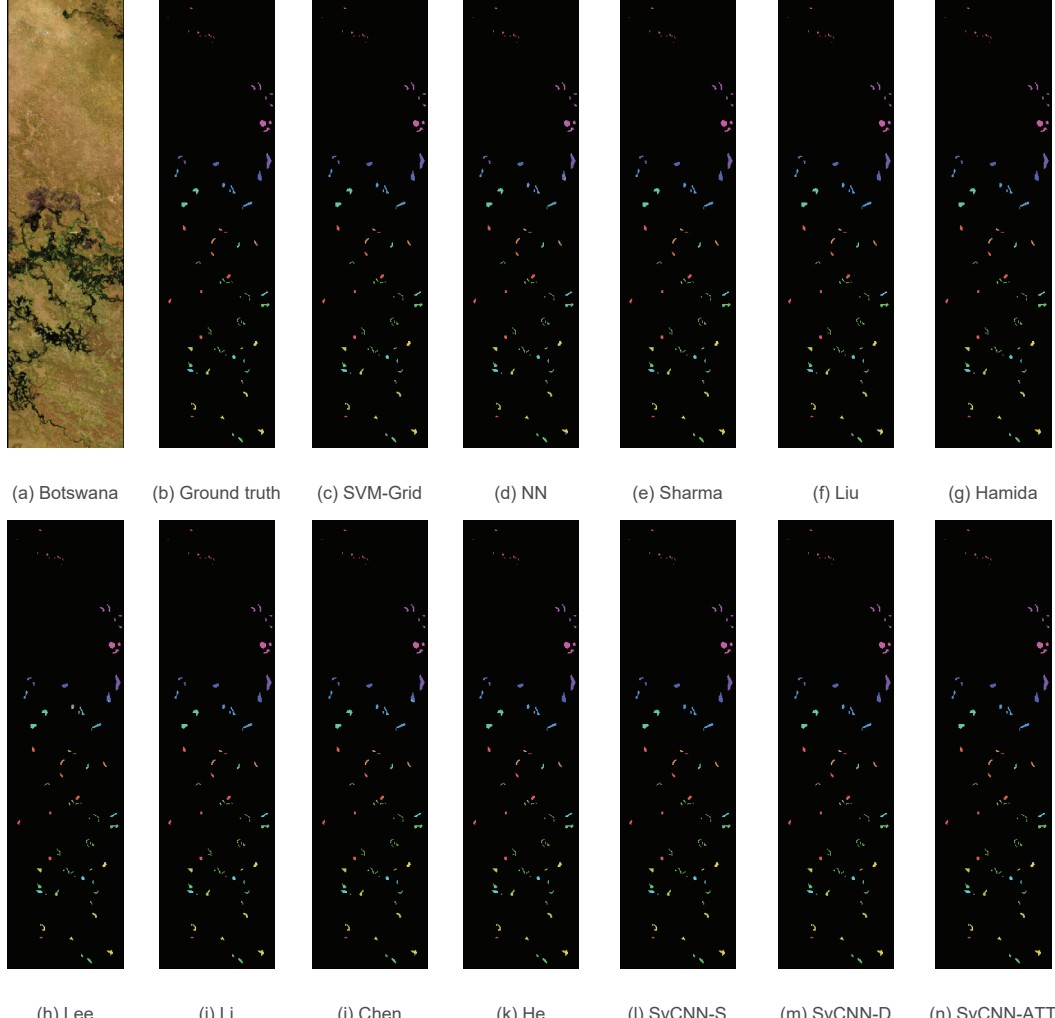

(a) Botswana  (b) Ground truth  (c) SVM-Grid  (d) NN  (e) Sharma  (f) Liu  (g) Hamida

(h) Lee  (i) Li  (j) Chen  (k) He  (l) SyCNN-S  (m) SyCNN-D  (n) SyCNN-ATT

**Figure 9.** Visualization of the experimental results based on Botswana: (**a**) Indian Pines, (**b**) Ground truth, (**c**) SVM-Grid, (**d**) NN, (**e**) Sharma, (**f**) Liu, (**g**) Hamida, (**h**) Lee, (**i**) Li, (**j**) Chen, (**k**) He, (**l**) SyCNN-S, (**m**) SyCNN-D, (**n**) SyCNN-ATT. It is observed that the outputs produced by our proposed models are quite close to the Ground truth.

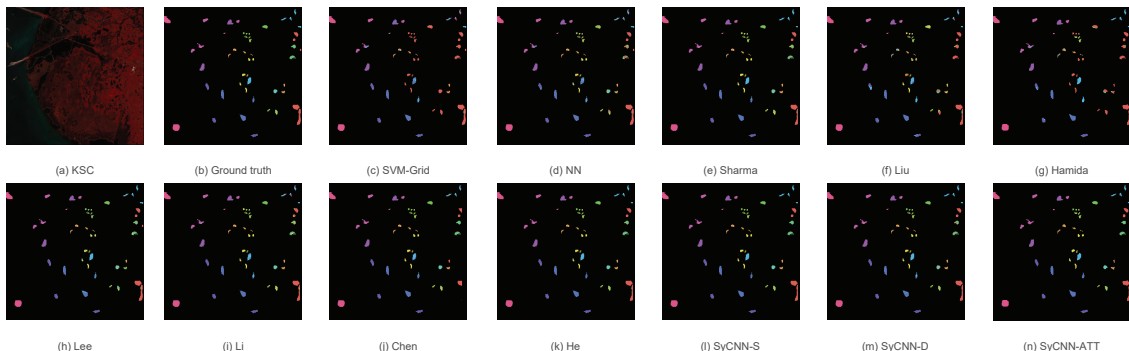

**Figure 10.** Visualization of the experimental results based on KSC: (**a**) Indian Pines, (**b**) Ground truth, (**c**) SVM-Grid, (**d**) NN, (**e**) Sharma, (**f**) Liu, (**g**) Hamida, (**h**) Lee, (**i**) Li, (**j**) Chen, (**k**) He, (**l**) SyCNN-S, (**m**) SyCNN-D, (**n**) SyCNN-ATT. It is observed that the outputs produced by our proposed models are quite close to the Ground truth.

#### 4.4.4. Effects on the Ratio of Training Examples

Apart from the adopted evaluation criteria, we also examined how the ratio of training examples could affect a model's classification performance. In this experiment, we varied the ratio of training examples from 10% to 70% and the experimental results based on all datasets are plotted in Figure 11. It is obvious that the performance of all models improves with the increasing number of training examples. This experimental result suggests that the deep neural network-based hyperspectral image classifiers generally need more training data to achieve better classification performance. It is also observed that the proposed approaches can successfully alleviate overfitting and outperform other approaches when only 10% training data was used. Based on the result of this experiment, we can conclude that the proposed three SyCNN-based methods can achieve good classification performance even a small training set is provided. Such a characteristic of the SyCNN-based methods is promising as the available training data for most real-world image classification applications is usually quite limited. It's more capable to handle spatial–spectral information and each part of the SyCNN also helps to improve performance. We note that the design of the SyCNN makes it possible to produce a promising result based on a small dataset.

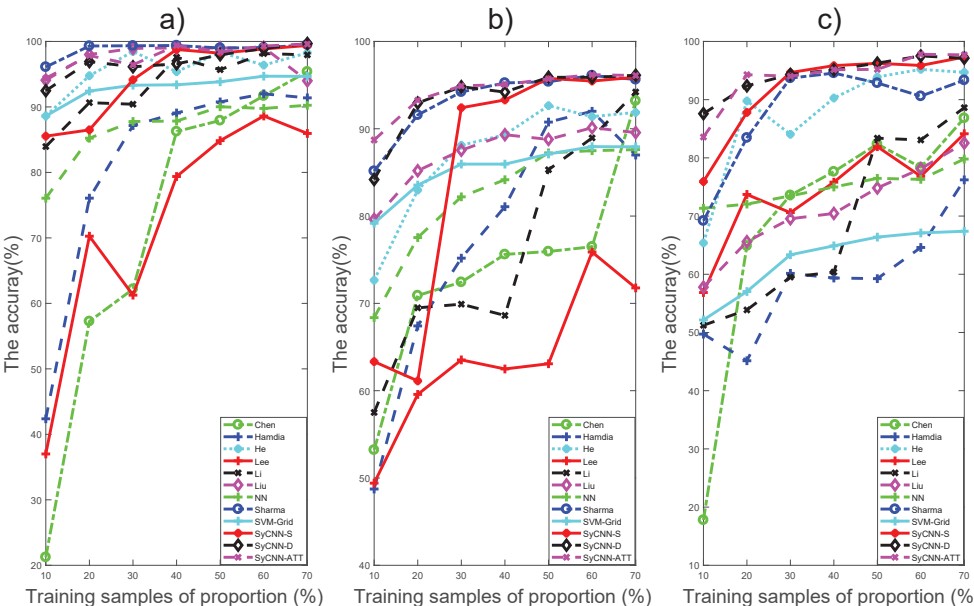

**Figure 11.** Visualization of the influence of training samples proportion for different methods based on the three datasets: (**a**) Indian Pines, (**b**) Botswana, (**c**) KSC. It can be observed that the results of our proposed models are very stable and better than the comparison methods.

### 4.4.5. Comparison of Parameters and Times of Different Methods

The comparison results about execution time of different models are reported in Table 7. The experiments are implemented on Botswana dataset. It could be seen that the proposed methods have a number of network parameters, including parameters of 2D CNNs, data interact module, and 3D CNNs. However, it does not significantly increase the model complexity. The execution time is not as high as Liu's. The possible reason is that the model has more operations such as convolutional operations. It also can be seen that the proposed methods outperform all methods.

**Table 7.** Comparison of parameters and times of all methods.

| Methods | Number of Parameters | Execution Time (s) | OA |
|---|---|---|---|
| SVM-Grid | 2.98 MB | - | 67.39 |
| NN | **65.52 MB** | 20.57 | 79.79 |
| Sharma | 5.9 KB | 25.86 | 93.35 |
| Liu | 4.08 MB | **8282.78** | 84.27 |
| Hamida | 472 KB | 46.88 | 76.22 |
| Lee | 1.22 MB | 128.29 | 84.14 |
| Li | 2.38 MB | 37.90 | 88.62 |
| Chen | 3.27 MB | 285.63 | 86.83 |
| He | 8.18 KB | 98.67 | 94.69 |
| SyCNN-S | 21.14 MB | 117.03 | 97.44 |
| SyCNN-D | 29.46 MB | 146.52 | 97.76 |
| SyCNN-ATT | 34.98 MB | 154.15 | **98.92** |

## 5. Conclusions

In this paper, we first propose a simple synergistic trained deep learning model, which is constructed by mixing 2D CNNs and 3D CNNs to extract deeper spatial–spectral features with fewer 2D/3D convolutions. We further present a deep SyCNN network for hyperspectral image classification, which introduces a data interaction module into the simple synergistic trained 2D/3D model. Experiment results show that our proposed deep SyCNN model has obtained a robust, good result and outperforms state-of-the-art methods for image classification. We also introduce 3D attention module into the deep SyCNN model to gain a state-of-the-art performance.

One explanation of the performance improvement of the SyCNN network could be that a deep network generally benefits by deepening the model. Another is based on the observation that there is stronger spectral redundancy than a spatial one in hyperspectral images. Thus, we propose a synergistic trained deep learning model that consists of 2D CNNs and 3D CNNs to further enhance the abstraction ability on the spatial domain and explore more useful information on the spectral domain. In the future, we will seek to design a optimal deeper SyCNN network for hyperspectral image classification. On the other hand, we will introduce a relation network into the SyCNN to learn the relationship between 2D and 3D output features, which is proposed to enhance the feature extraction process.

**Author Contributions:** X.Y., this paper first author, write the manuscript and correct it. X.Z., the Correspondence author and correct this paper. Y.Y., R.Y.K.L., S.L., X.L. and X.H., these are co-authors and also correct this paper. X.Z., Y.Y., R.Y.K.L., X.L. and X.H. also provide the funding to help me. All authors have read and agreed to the published version of the manuscript.

**Funding:** This work was partially supported by the National Key R&D Program of China, 2018YFB0504900, 2018YFB0504905 and the Shenzhen Science and Technology Program under Grant JCYJ20170811160212033, Grant JCYJ201603301639000579, Grant JCYJ20180507183823045, and Grant JCYJ20170413105929681. Zhang's work was supported by the National Science Foundation of China under Grant No.61872108, Shenzhen Science and Technology Program under Grant No.201708113000098. Lau's work was supported in part by the Research Grant Council of the Hong Kong SAR (Projects: CityU 11502115 and CityU 11525716), the National Natural Science Foundation of China (NSFC) Basic Research Program (Project 71671155), and CityU Shenzhen Research Institute. X. Huang was supported in part by Natural Science Foundation of Jiangxi Province under Grant No.20192ACBL21006.

**Acknowledgments:** This work was partially supported by the National Key R&D Program of China, 2018YFB0504900, 2018YFB0504905 and the Shenzhen Science and Technology Program under Grant JCYJ20170811160212033, Grant JCYJ20160330163900579, Grant JCYJ20180507183823045, and Grant JCYJ20170413105929681. Zhang's work was supported by the National Science Foundation of China under Grant No.61872108, Shenzhen Science and Technology Program under Grant No.201708113000098. Lau's work was supported in part by the Research Grant Council of the Hong Kong SAR (Projects: CityU 11502115 and CityU 11525716), the National Natural Science Foundation of China (NSFC) Basic Research Program (Project 71671155), and CityU Shenzhen Research Institute. X. Huang was supported in part by Natural Science Foundation of Jiangxi Province under Grant No.20192ACBL21006.

**Conflicts of Interest:** The authors declare no conflicts of interest.

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
