# Peer review of "Synergistic 2D/3D Convolutional Neural Network for Hyperspectral Image Classification"

_remotesensing, doi:10.3390/rs12122033_

Round 1
Reviewer 1 Report
The paper proposed a novel method of classifying data sampled from hyperspectral satellite imagery. The most important contribution of the paper is the feature fusion module mixing learned features from each stage of 2D convolution layers and 3D convolution layers.
The *-D notation could be confusing that 2D conv layers only focus on spatial dimensions and do not learn the spectrum of the imagery. On the contrary, both 3D and 2D convolutions integrate spectrum information.
From my point of view, it is clearer to emphasize the spectral locality of 3D convolution layers and the spectral holistic of 2D convolution layers. In other words, 2D ops combine ALL the spectral channels at once at each step, while 2D ops take into account a handful of them in each convolution op. This is somewhat similar to enforcing group sparsity into the spectral dimension. The superior performance of the network henceforth is the result of combining dense and sparse feature learning. Such a viewpoint could expose interesting justification of why the network performs better than related works. This opinion however is a matter of subjectiveness from the reviewer, thus not criticizing any particular part of this work. Of course, if the authors could provide more theoretical based explanations into the paper, scientific quality would be much better.
There are Figure 6 and Figure 7 need a rework. It is very difficult to differentiate the details. Hence I suggest to find another way to represent the figure. For example, to localize each blob into a patch and arrange those ones in a matrix, avoiding showing large empty space which is useless. Error / difference map between ground truth and the predicted result is also better to be viewed.
The authors emphasize the ability of the network to perform well under scarce data; however, the paper is lack of experiment description of how the network copes against overfitting under the lack of training data. An in-depth explanation about optimization and training steps are very appreciated in order to make the work reproducible.
Overall this is good work. I strongly encourage the authors to improve representation in certain parts to make the paper even better.
Author Response
We would like to thank the Reviewer for the helpful comments, and we have tried our best to revise the manuscript according to your comments.
- From my point of view, it is clearer to emphasize the spectral locality of 3D convolution layers and the spectral holistic of 2D convolution layers. In other words, 2D ops combine ALL the spectral channels at once at each step, while 2D ops take into account a handful of them in each convolution op. This is somewhat similar to enforcing group sparsity into the spectral dimension. The superior performance of the network henceforth is the result of combining dense and sparse feature learning. Such a viewpoint could expose interesting justification of why the network performs better than related works. This opinion however is a matter of subjectiveness from the reviewer, thus not criticizing any particular part of this work. Of course, if the authors could provide more theoretical based explanations into the paper, scientific quality would be much better.
Response: First, we would like thank you for your valuable time. The proposed SyCNNs not only combine dense and sparse features, but also generate sufficient features for the next 3D convolutional operation, and more spatial-spectral features for the next 2D convolutional operation. In fact, with the data interaction module, the proposed SyCNNs make full use of the spatial-spectral fusion information.
2. There are Figure 6 and Figure 7 need a rework. It is very difficult to differentiate the details. Hence I suggest to find another way to represent the figure. For example, to localize each blob into a patch and arrange those ones in a matrix, avoiding showing large empty space which is useless. Error / difference map between ground truth and the predicted result is also better to be viewed.
Response: We have updated the Figure 6 and Figure 7 on Page 14.
3. The authors emphasize the ability of the network to perform well under scarce data; however, the paper is lack of experiment description of how the network copes against overfitting under the lack of training data. An in-depth explanation about optimization and training steps are very appreciated in order to make the work reproducible.
Response: We have added explanations about how the network copes with overfitting problems. In fact, the Botswana Scene dataset is a scarce data, and we copy the content in Page 13 as below.
“Botswana Scene dataset only has 3248 samples, and the labels are unbalanced. Similarly, the proposed approaches can achieve the best classification performance. Due to the power of the data interaction module, the proposed SyCNN-D could extract the deeper and richer features from the hyperspectral images. The 2D convolutional operation could extract much more spatial information by treating the whole spectral information as channels, but it loses the relationship between the different spectral bands. Although, the 3D convolutional operation cloud extract much more spatial-spectral fusion information by utilizing a stable step over spectral information, it could only combine a few spectral information with a fixed size and need more training samples for the next operation. In the data interaction module, the 2D extracted features could be projected into 3D features and the projected 3D features are fusion in the 3D extracted features. Thus, the new 3D features combine more spectral information and make full use of spatial information. In addition, the new 3D features are enough to train the next 3D convolutional operation. So as the new 2D features that have more spectral information and enough to train the next 2D convolutional operation.”
We also added some explanations about the experiments which are reported in Page 7.
“Note that the BN-inception [38] and the ReLU function [29] are applied after each convolutional block in the proposed model, which is used to address the overfitting problem caused by the scarce data and limited training samples in hyperspectral images.”

Reviewer 2 Report
The authors present a hybrid model that combines 2D and 3D CNNs and a data interaction module that fuses spectral and spatial hyperspectral information for hyperspectral image classification.
The topic is interesting and the methodology is well described. Just as a suggestion, the authors could discuss a little more about the advantages of the proposal. Spatial information is already included in 3D CNNs, so one of the benefits of separating the two models is that spatial resolution can be increased in the 2D CNN. If I understand correctly, the data interaction consists basically of concatenating the 2D and 3D outputs. Have you considered other forms of data fusing?
Finally, it would be interesting to include some results comparing the execution time and the number of weights of each model.
Author Response
We sincerely thank the Reviewer for the kind suggestions and we have tried our best to address these comments.
- The topic is interesting and the methodology is well described. Just as a suggestion, the authors could discuss a little more about the advantages of the proposal. Spatial information is already included in 3D CNNs, so one of the benefits of separating the two models is that spatial resolution can be increased in the 2D CNN. If I understand correctly, the data interaction consists basically of concatenating the 2D and 3D outputs. Have you considered other forms of data fusing?
Response: For the data interaction module, the benefits are two-fold. One is that the spatial resolution can be increased in the 3D CNN, and the relationship between different spectral bands can be captured by the 2D CNN. Another is that there have enough training samples to train the next 3D and 2D convolutional operations.
Due to the characteristically of hyperspectral images, we only consider fusing the features of 2D and 3D convolution operations. We will try other types of data fusing process, such as 1D fusion.
- Finally, it would be interesting to include some results comparing the execution time and the number of weights of each model.
Response: We have added the results about the execution time and the number of weights. More detailed results are reported in Page 16, and we copy the content as below
“The comparison results about the execution time of different models are reported in Table.7. The experiments are implemented on the Botswana dataset. It could be seen that the proposed methods have a number of network parameters, including parameters of 2D CNNs, data interact module, and 3D CNNs. However, it does not significantly increase the model complexity. The execution time is not as high as Liu's. The possible reason is that the model has more operations such as convolutional operations. It also can be seen that the proposed methods outperform all methods.”
